# Improving Interaction at Polymer–Filler Interface: The Efficacy of Wrinkle Texture

**DOI:** 10.3390/nano10020208

**Published:** 2020-01-25

**Authors:** Pietro Russo, Virginia Venezia, Fabiana Tescione, Joshua Avossa, Giuseppina Luciani, Brigida Silvestri, Aniello Costantini

**Affiliations:** 1Institute for Polymers, Composites and Biomaterials, National Research Council, via Campi Flegrei 34, 80078 Pozzuoli-Naples, Italy; pietro.russo@unina.it; 2Department of Chemical, Materials and Production Engineering, University of Naples “Federico II”, p.le V. Tecchio 80, 80125 Naples, Italy; virginia.venezia@libero.it (V.V.); luciani@unina.it (G.L.); anicosta@unina.it (A.C.); 3Institute for Polymers, Composites and Biomaterials, National Research Council, Portici, 80055 Naples, Italy; fabiana.tescione@libero.it; 4Institute of Atmospheric Pollution Research-National Research Council (IIA-CNR), Research Area of Rome 1, via Salaria Km 29,300, 00016 Monterotondo, Italy; joshua.avossa@iia.cnr.it

**Keywords:** nanocomposites, Stöber silica nanoparticles, wrinkled nanoparticles, PLA

## Abstract

One of the main issues in preparing polymer-based nanocomposites with effective properties is to achieve a good dispersion of the nanoparticles into the matrix. Chemical interfacial modifications by specific coupling agents represents a good way to reach this objective. Actually, time consuming compatibilization procedures strongly compromise the sustainability of these strategies. In this study, the role of particles’ architectures in their dispersion into a poly-lactic acid matrix and their subsequent influences on physical-chemical properties of the obtained nanocomposites were investigated. Two kinds of silica nanoparticles, “smooth” and “wrinkled,” with different surface areas (≈30 and ≈600 m^2^/g respectively) were synthesized through a modified Stöber method and used, without any chemical surface pre-treatments, as fillers to produce poly-lactic acid based nanocomposites. The key role played by wrinkled texture in modifying the physical interaction at the polymer-filler interface and in driving composite properties, was investigated and reflected in the final bulk properties. Detailed investigations revealed the presence of wrinkled nanoparticles, leading to (i) an enormous increase of the chain relaxation time, by almost 30 times compared to the neat PLA matrix; (ii) intensification of the shear-thinning behavior at low shear-rates; and (iii) slightly slower thermal degradation of polylactic acid.

## 1. Introduction

Poly-lactic acid (PLA) is one of the most promising “green plastics.” It is a bio-compostable and bio-degradable linear aliphatic polyester, typically derived from renewable sources, such as corn, starch and sugar beet. With its universal applications in different fields, PLA is considered one of the most attractive candidates to replace petroleum-based polymers. However, it still displays some serious drawbacks, such as limited melt strength, poor toughness and a lack of functional groups, and different modification approaches have been proposed to tune desirable performances [1,2,3].

Currently, the inclusion of organic or inorganic reinforcements is widely considered, especially if they have nanometric dimensions, since the high specific surface area of the nanoparticles permits a guarantee of obtaining products with improved properties even in the presence of relatively low quantities of the same [4,5].

To date, different types of inorganic fillers have been used for the production of PLA-based nanocomposites such as carbon nanotubes (CNTs) [6,7,8], graphene [9,10,11], natural rubber [12,13,14], silicates [15,16] and cellulose [17,18,19]. Among the different types of inorganic nanomaterials, silica particles have been receiving growing attention by the scientific community as an outstanding choice for reinforcement materials due to their low price, high strength modulus and functional versatility [20]. Moreover, the sol-gel Stöber method allows fine tuning of size, size distribution, shape and surface chemistry [21,22,23,24], thereby influencing properties of nanocomposites, including melt behavior, thermal stability and mechanical and/or rheological features [25,26].

However, the high hydrophilic nature of sol-gel derived silica particles and the hydrophobic PLA make the compatibility between the two phases low. Therefore, a good dispersion is not easily achieved: the different nature of the polymer matrix with respect to the nanofiller is responsible for its strong tendency to form aggregates. This heavily compromises the performance of obtained composites, strongly related to the level of the filler dispersion actually achieved [2,27].

In order to achieve excellent dispersion of the filler, many strategies have already been employed. These include conventional synthetic approaches such as melt-extrusion processes [28], co-extrusion and solution casting [29,30,31,32]; the inclusion of interfacial modifications by specific coupling agents, including oleic acid, rubbers, L-lactic acid oligomers and 2-methacryloyloxyethyl isocyanate [33]; and/or surface pre-treatments of the reinforcing phase [34,35].

Actually, the use of compatibilizer compounds allows one to obtain a good dispersion of the fillers in the matrix, often resulting in time consuming procedures; therefore, research efforts are focused on different routes able to face this challenge.

Mesoporous silica particles have served as an effective reinforcement filler to enhance properties of a wide range of different polymers [36]. The diffusion of the polymeric matrix into nanometric-sized pores of nanofillers enhances the interfacial interactions and strongly suppresses aggregation [37,38]. Among various structures of mesoporous silica nanoparticles, wrinkled silica nanoparticles have recently grown in popularity because they provide excellent pore accessibility. These nanoparticles exhibit an open pore structure, where the radial pore channel size increases going from the interior to the surface, thereby enhancing the accessibility of macromolecules (67.5 kDa) inside the pores [39].

In our previous study [26], we proposed a novel, straightforward strategy, to produce PBT/SiO_2_ nanocomposites by using the same polymeric components as endogenous coupling agents. This approach hinders the typical irreversible aggregation of silica particles and at the same time allows one to obtain blends at higher filler content without any coupling agent. 

The objectives of the present study were to investigate the role of particles’ architectures on their dispersion into PLA matrix and the subsequent influences on physical-chemical properties of the obtained nanocomposites.

To achieve our aims, two different types of silica nanoparticles, “smooth” and “wrinkled,” of 250 nm diameters, were produced through a modified Stöber method and used to fill a commercial grade PLA. Furthermore, no chemical surface pre-treatments were carried out to improve the compatibility at the interface between the two phases.

Melt-mixing methodology was used to synthesize nanocomposites with 3 wt% of both smooth and wrinkled nanoparticles, and textural features of all synthesizes sample were carefully investigated.

## 2. Experimental Section

### 2.1. Materials

A commercial grade poly(lactic acid) was supplied by Nature Work under the trade code 7001D (density: 1.24 g/cm^3^; Tg: 57.1 °C; Tm: 154.0 °C; $\bar{M_{w}:}$ 113.000 g/mol; polydisperity index: 1.59) and used as the matrix of the nanocomposites. Tetraethoxysilane (TEOS), ammonium hydroxide, acetone, ethanol, chloroform, urea, ciclohexane, hexadecyltrimethylammonium bromide and 2-propanol were purchased from Sigma-Aldrich, Milan, Italy, and used as received.

### 2.2. Synthesis of SiO_2_ Nanoparticles

Two different kinds of silica nanoparticles of uniform sizes were prepared. The former, characterized by a smooth surface morphology, were produced according to the Stöber method [40], and denoted as SiO_2__s NPs. The latter nanoparticles, characterized by a wrinkled structure, were prepared by following the procedure of Yang et al. [41]. Briefly, a water solution (60 mL) of CTAB (2.0 g) and urea (1.2 g) was prepared. Subsequently, cyclohexane (60 mL) and isopropyl alcohol (1.84 mL) were added to the solution, and then, the TEOS (5.0 mL) was added dropwise. After continuous stirring for 30 min at room temperature, the reaction mixture was heated up to 70 °C and kept for 16 h at this temperature in a closed system to avoid the evaporation of the solvent. The product was obtained by centrifugation and washed with ethanol thrice. The surfactant’s removal was ensured by rinsing the particles, hereinafter referred to as SiO_2__w NPs, in 500 mL of ethanol and 40 mL of acid chloride solution for 16 h.

### 2.3. Synthesis of PLA-Modified SiO_2_ Nanoparticles

Both SiO_2__s and SiO_2__w nanoparticles were resuspended in a solution of PLA previously dissolved in chloroform/acetone, 2:1 volume ratio, in order to have a 40 wt% of silica nanoparticles into the final PLA-SiO_2_ mixture. The two mixtures were dried overnight at 50 °C under vacuum, and the obtained dried powders are indicated in the following text as PLA/SiO_2__s NPs and PLA/SiO_2__w NPs respectively.

### 2.4. Synthesis of PLA-SiO_2_ Nanocomposites

Appropriate amounts of both PLA/SiO_2__s and PLA/SiO_2__w nanoparticles were incorporated into pure PLA to obtain a final concentration of 3.0 wt% silica nanoparticles. These materials are denoted PLA/SiO_2__s NCs and PLA/SiO_2__w NCs. Nanocomposites with different amounts of only wrinkle nanoparticles (0.5, 1 wt%) were also produced, and are denoted PLA/SiO_2__w0.5 NCs and PLA/SiO_2__w1 NCs.

Nanocomposites were prepared using a co-rotating fully intermeshing twin-screw mini-extruder (DSM Research, Sittard, The Netherlands). This apparatus has an internal capacity of about 4 cm^3^ and it is characterized by the presence of a recirculating channel. In detail, raw materials, pre-dried at 90 °C overnight, were compounded at 180 °C, with the screw speed at 120 rpm and employing a residence time of 3 min.

Neat PLA processed under the same conditions was considered the reference material.

### 2.5. Characterization Techniques

Transmission electronic microscopy (TEM) analysis was performed using a FEI Tecnai G12 Spirit-Twin (LaB6 source) equipped with a FEI Eagle 4k CCD camera, operating with an acceleration voltage of 120 kV.

Morphological investigation of all synthesizes samples was carried out by adding nanoparticles in ethanol solution. A drop of the obtained suspension was placed one side of the transparent polymer coated 200 mesh copper grid and a carbon layer was deposited on the samples surface.

The morphological investigation of nanocomposites was performed using a Leica UC6/FC6 ultramicrotome at 2160 8C in order to obtain ultra-thin sections (120 nm), and TEM images were obtained by using a Mega View camera connected to PHILIPS EM208S microscope.

A scanning electron microscopy (SEM) investigation was performed in order to investigate the surface morphology of the prepared nanocomposites. A FEI Quanta 200F microscope was used and a gold layer was deposited on all sample surfaces.

Fourier transform infrared (FT-IR) transmittance spectra were obtained by using a Nexus FT- IR spectrometer connected to DTGS KBr detector. Pellets of 200 mg were obtained by dispersing 0.5 mg of the nanoparticles in KBr, and all spectra were recorded in the 4000–400 cm^−1^ range, with a 2 cm^−1^ value of resolution, and KBr was used as the blank.

The textural features of wrinkled silica nanoparticles were highlighted by N_2_ adsorption at −196 °C with a Quantachrome Autosorb 1-C, after degassing for 4 h at 150 °C. Specific surface area was evaluated by using the Brunauer–Emmett–Teller (BET) method. Barrett–Joyner Halenda (BJH) adsorption method and Dubinin–Astakov (DA) method were used to evaluate mesopore and micropores size distributions respectively.

Structural characterization of wrinkled SiO_2_ particles was performed through small angle X-ray scattering SAXS analyses by using an Anton Paar SAXSess camera equipped with a 2D imaging plate detector. CuKα X-Rays with 1.5418 Å wavelength were generated by a Philips PW3830 sealed tube source (40 kV, 50 mA) and slit collimated. The spectra were collected in transmission mode for 15 min to assure a good signal/noise ratio. All scattering data were corrected for background and normalized for the primary beam intensity [42,43]. The Porod constant and desmearing effect were also corrected to exclude from SAXS profiles the inelastic scattering.

The analysed q-range was 0.12–5 nm^−1^, where q is the magnitude of the scattering vector (see Equation (1)):Q = (4 π/λ) sin θ,(1)
in which θ is the scattering angle and λ is the wavelength of the Cu Kα radiation (0.1542 nm). The SAXS spectrum is reported as a corrected plot (i.e., I⋅q^2^ versus q).

TG analysis was performed in a TG apparatus, TGA Q5000 TA Instruments. All investigated samples were heated to 700 °C at a rate of 20 °C/min under nitrogen atmosphere.

Calorimetric analyses were performed were performed with a Mettler Toledocalorimeter (model DSC1). The tests were carried out under a nitrogen atmosphere. The samples were heated from 70 °C to 180 °C at a rate of 10 °C/min.

Rheological measurements were conducted under a nitrogen atmosphere using a stress-controlled ARES rheometer operating in dynamic oscillatory mode with a parallel plate geometry (diameter 25 mm, gap: 0.7 mm). All materials, preliminarily dried at 60 °C overnight, were compression moulded in disks (25 mm diameter, 1 mm thickness) before each experiment. Tests were performed within the linear viscoelastic domain previously identified by strain-sweep measurement. In particular, complex modulus component (G’, G”) and complex viscosity (η*) data were collected by subjecting the materials to dynamic tests under a strain of 1% over a frequency range of 0.01–100 Hz at 180 °C.

All evaluated parameters are reported as a function of the oscillatory frequency (ω).

## 3. Results and Discussion

Figure 1 shows the TEM images of SiO_2__s (A, B) and SiO_2__w (C, D) nanoparticles at different magnifications. Both nanoparticles (NPs) appeared well separated and showed spherical morphology with a narrow size distribution of about 250–300 nm. Furthermore, SiO_2__w nanoparticles (Figure 1C,D) exhibited clear fibrous morphology and radial wrinkle structure; the wrinkles spread uniformly in all directions, forming central-radial pores that widened radially outward.

The FT-IR spectrum of wrinkled NPs (Appendix A) showed the typical absorption band of silica gel. The peak at 1100 cm^−1^ was attributed to Si–O–Si stretching vibration modes in SiO_4_ units, that at 470 cm^−1^ is usually assigned to Si–O–Si bending, whereas the peak at 800 cm^−1^ was attributed to Si–O–Si bond vibration between two adjacent tetrahedra. Furthermore, the peak at 950 cm^−1^ was assigned to Si–O terminal nonbridging vibration.

The porosity of both nanoparticles was assessed by means of N_2_ adsorption/desorption measurements (Appendix A). A non-porous structure with a surface area of about 30 m^2^/g was obtained for SIO_2__s NP_S_ (Appendix A). A type IV isotherm, typical for mesoporous materials, was obtained in the case of wrinkled NPs (Appendix A). The surface area was about 580 m^2^/g, while the total pore volume was 1.7 cm^3^/g, in accordance with previously obtained results [39]. The pore size distribution (Appendix A) indicated the presence of mesopores in 5–50 nm and 2–4 nm ranges, suggesting a mesoporous structure in addition to wrinkles. Moreover, additional microporosity was found in the range of 1–2 nm, indicating a hierarchical porous structure [44].

In order to investigate the ordered mesoporous structure, the SAXS scattering investigation of SiO_2__w nanoparticles were performed. The XRD pattern, reported in Figure 2, showed a more defined peak related to the (100) diffraction plane and two broader and less intense peaks assigned to (110) and (200) planes usually attributed to 2D hexagonal pore array (P6 mm symmetry). The not-well-resolved peaks at (110) and (200) planes can be ascribed to short range ordered 2D hexagonal pore array [45,46,47].

After the functionalization with PLA (Figure 3), the nanoparticles showed similar morphologies and sizes of bare NPs. However, both PLA/SiO_2__s (Figure 3A) and PLA/SiO_2__w (Figure 3B) nanoparticles appeared to be composed of a dense dark silica core and a gray contrast halo, indicating the presence of a less dense component due to the presence of PLA. Furthermore, hierarchical wrinkled architecture can hardly be appreciated in Figure 3B, suggesting that the polymer phase filled the pores of nanoparticles.

Both smooth and wrinkled nanoparticles appear homogeneously dispersed in the polymeric matrix, as highlighted from the SEM micrographs (Figure 4) of PLA/SiO_2__s (Figure 4A) and PLA/SiO_2__w (Figure 4B) nanocomposites. The corresponding TEM images of ultra-thin sections of both nanocomposites at low magnification, reported as inserts of Figure 4, indicated the achievement of an almost similar dispersion regardless of the type of nanoparticles included.

Figure 5A reports the thermogravimetric (TG) analysis of pure PLA, PLA/SiO_2__s and PLA/SiO_2__w nanocomposites. The TG curves show similar profiles of one step degradation at about 350 °C due to the thermal decomposition of pure PLA.

The 3 wt% residues at 700 °C, related to the inorganic fillers, agree with the nominal composition. Furthermore, thermal degradation is slowed by the inclusion of silica nanoparticles, as confirmed by the slight shift of the derived signal towards higher temperatures for nanocomposite materials (see Figure 5B). The maximum decomposition temperatures of PLA and PLA/SiO_2__w nanocomposites were 370 and 380 °C, respectively. In other words, the presence of wrinkled nanoparticles seems to slightly slow down the typical phenomena of thermal degradation of polylactic acid.

In fact, the hierarchical porous structure of wrinkled silica was expected to allow mass transport within an inorganic network, enhancing molecular diffusion into the pores, allowing pore filling by the polymer (see Figure 3B). Immobilization of the polymer chains inside the NPs channels and the interaction between the two components ultimately lead to a delayed release of the degradation product from the sample, resulting in different thermal degradation behavior with respect to bare PLA and PLA/SiO_2__s nanocomposites.

Figure 6 shows the Differential Scanning Calorimetry (DSC) thermograms of all studied materials recorded during the 1st (Figure 6A) and the 2nd (Figure 6B) heating steps. Processing these curves indicated that the cold-crystallization temperature of bare PLA (112 °C) was slightly increased by the included silica nanoparticles, and with the inclusion of both silica particles, slightly shifted to lower values, probably due to nucleating effects induced by the NPs. Furthermore, both “smooth” and “wrinkled” silica NPs did not significantly influence the thermal behavior of the samples.

Rheological tests are widely used to estimate the actual level of filler dispersion in nanocomposites: An adequate dispersion is responsible for the formation of percolative network structures affecting the viscoelastic behavior of polymer compounds at low frequencies [3,18].

Figure 7 shows the complex viscosity of the investigated materials as a function of the frequency. Clearly, the terminal zone of the neat PLA curve is characterized by an initial plateau with a zero shear viscosity approximately equal to $4\cdot 10^{2}\;\mathrm{Pa}\cdot s$. This low-frequency region is significantly modified in the presence of the nanoparticles by the appearance of a shear thinning behavior just evident in presence of Stӧber SiO_2_ nanoparticles, but much more pronounced for the sample containing the same amount of the wrinkled filler. This effect is usually ascribed to flow restrictions of polymer chains in the molten state due to the organization of included nanoparticles in more or less complex three-dimensional meso-structures between filler particles and polymer chains. The phenomenon, best known as solid-like behavior, is a result of particle-particle and/or particle-matrix interactions and it is a sign of the achievement of a so-called rheological percolation threshold [48,49,50,51].

In general, the magnitude of this effect increases as the dispersion level of the nanoparticles included increases. However, considering that in our case, as evidenced by the morphological analysis previously discussed, the dispersion obtained is almost independent of the surface characteristics of the added particles, the greater intensity of the shear thinning effect highlighted for samples containing wrinkled particles can be attributed to the occurrence of more marked physical interactions among these particles and the surrounding polymeric matrix.

The frequency dependence of the storage modulus (G’), representing the elastic response of the considered materials, is shown in Figure 8.

In this case, as is better highlighted in Table 1, the terminal behavior of neat PLA is characterized by a scaling factor (n) of G’ versus ω approximately equal to 1.32. The magnitude of this value, lower than the theoretical one of 2, consistent with the linear viscoelastic theory, can be ascribed to the polydispersity and stereoregularity of the considered PLA matrix [52].

Analogously, the inclusion of fillers mainly affected the low-frequency section of the G’ curves with a relevant reduction of the terminal slope due to the formation of network structures particularly enhanced in presence of wrinkled SiO_2_ nanoparticles (*n* = 0.074).

These considerations, partly attributable to expected weak hydrogen bonding interactions between the silanol groups (Si-OH) present on the surface of the filler particles and the carboxyl groups (C=O) of the matrix chains [53], are also widely supported by physical issues. In fact, evaluations of the so-called relaxation times of the polymer chains at 0.01 Hz according to the simple equation [54],
(2)$$\lambda =\frac{G^{\prime }}{\eta^{*}\omega^{2}},$$
demonstrated that this parameter not only increases in presence of silica particles, but that it is especially significantly influenced by physical features of the surface of included filler (see Table 2).

In particular, with the same silica content, the Stӧber particles induce an increase of the chain relaxation time by almost five times compared to the neat hosting matrix, while the wrinkled ones generate an enormous increase of the same parameters from about 40 s for the PLA to approximately 1290 s for the compound.

However, all cited effects faded at high oscillation frequencies, where the behavior of the matrix is dominant and the influence of the filler-matrix network becomes less incisive.

Figure 9 compares representative loss modulus curves (G”) as a function of the frequency for the investigated materials. In all cases, the trend appears monotonically increasing with higher G’’ values for composite materials compared to neat PLA. In particular, the highest values of this parameter are characteristic of the composite containing wrinkled SiO_2_ particles.

Nanocomposites with different amounts of wrinkle nanoparticles (0.5, 1 wt%) were also produced in order to assess the lowest filler amount allowing for significant enhancements in dynamical rheological properties. Representative curves of the storage modulus (G’) and loss modulus (G”) versus frequency (ω) are shown in Figure 10 for PLA based materials including various contents of wrinkled SiO_2_ nanoparticles. In particular, regarding G’ (Figure 10A), the liquid-like trend of the neat PLA fades at low frequencies where the slope of the curve progressively decreases up to the achievement of a plateau, as the filler content increases. When G’ is independent of the oscillation frequency (solid-like behaviour), the corresponding filler content represents the so-called rheological percolation threshold for the system under examination. On the other hand, at high frequencies (≥1 rad/s), although the presence of the filler translates the curves of G’ proportionally upwards with respect to that of the pure matrix, no variations in the slopes of the curves occur. In this region, the viscoelastic flow of the matrix becomes predominant with respect to the filler–matrix and filler–filler interactions responsible for the liquid–solid like transition described above.

As far as the loss modulus G’’ curves are concerned (Figure 10B), there is usually a progressive upward translation of a curve as the filler content increases, and all the investigated materials showed a similar trend.

Representative complex viscosity η* versus frequency (ω) curves for the examined materials are compared in Figure 11.

In this case, two regions can be distinguished to better describe the picture. At relatively low frequencies (≤1 rad/s), a drift of the curve more becoming more pronounced as the content of silica nanoparticles grows is evident: the increase in viscosity is due to the formation of entanglements between the polymer chains and the included particles which hinder the flow of the melt.

At medium oscillation frequencies (1/10 rad/s), the complex viscosity of the melt remains almost unchanged for relatively low filler contents (1 wt%), but a significant increase of the same parameter is detected for formulations containing 3 wt% of wrinkled SiO_2_ nanoparticles.

At high frequencies (≥10 rad/s) polymer chains are fully oriented in the flow direction and a typical shear-thinning behaviour is always observed with more and more filler concentration effects fading.

These results show that a sensitive improvement in the dynamical rheological properties can be achieved only with a 3 wt% of wrinkle particles.

Finally, Figure 12 shows the angular frequency dependence of the loss factor (Tan δ) for the same materials. In more details, a decreasing trend, typical of viscoelastic fluids, characterizes neat PLA.

This behavior is influenced by the inclusion of particles, which induce a reduction of this parameter, especially in the low frequency region. This effect, once again particularly pronounced in the case of the material filled with wrinkled particles, indicates an increased melt elasticity and can be attributed to the formation of physical networks between the SiO_2_ particles and surrounding polymer chains.

In light of the above considerations, it is possible to state that the peculiar morphological structure of the wrinkled nanoparticles with respect to the Stӧber ones, in addition to ensuring, at the same concentration, a greater physical interaction with the surrounding polymeric matrix, can also favor more effective particle–particle interactions. These latter promoted the formation of percolative network structures that hinder the matrix flow and enhance the elastic behavior.

## 4. Conclusions

“Smooth” and “wrinkled” silica particles were used as fillers to produce PLA based nanocomposites through a melt-mixing procedure. The influence of silica’s texture on the properties of the synthesized samples was investigated by rheological experiments.

The adopted synthesis methodology, based on the use of PLA as endogenous coupling agent, allowed a homogeneous distribution of filler into the polymeric matrix for both nanocomposites.

The peculiar wrinkled structure ensured a greater physical interaction with the surrounding polymeric matrix and favored more effective particle–particle interactions. The consequent percolative network structures hindered the matrix flow, ultimately leading to a notable enhancement in the elastic behavior. Furthermore, the surface roughness of porous particles had polymer significantly interlocked within pores, promoted by the huge contact area between filler and matrix. This effect, among others, slightly shifts the temperature at which the maximum rate of PLA thermal degradation occurs.

Overall, this study discloses the key role played by surface topology and texture in driving material properties, providing proof of concept for a design approach to tune technological properties of nanocomposites.

## Figures and Tables

**Figure 1 nanomaterials-10-00208-f001:**
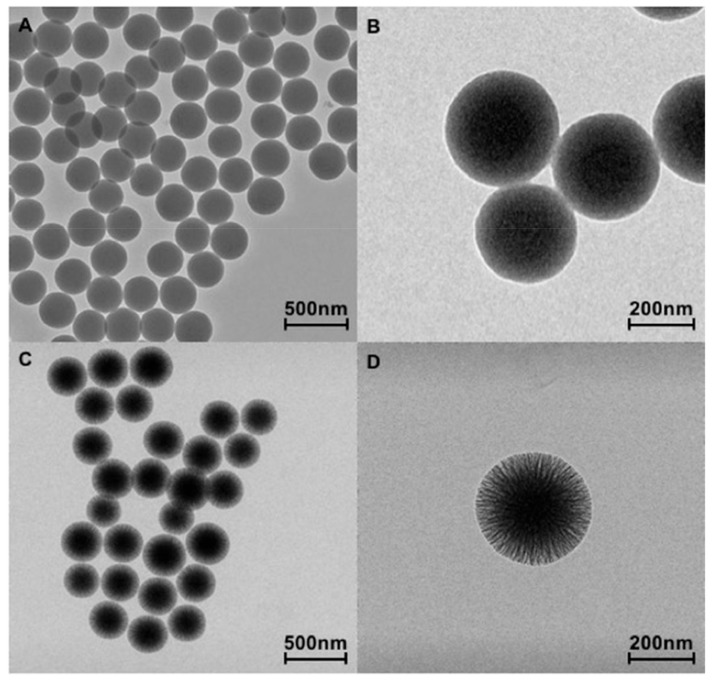
TEM images of SiO_2__s NPs (smooth) (**A**,**B**) and SiO_2__w (**C**,**D**) nanoparticles (wrinkled) at different magnifications.

**Figure 2 nanomaterials-10-00208-f002:**
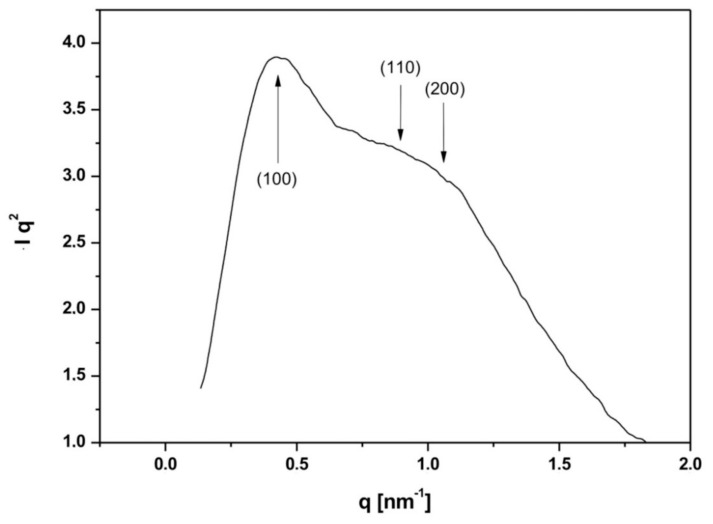
SAXS diffractogram of SiO_2__w NPs.

**Figure 3 nanomaterials-10-00208-f003:**
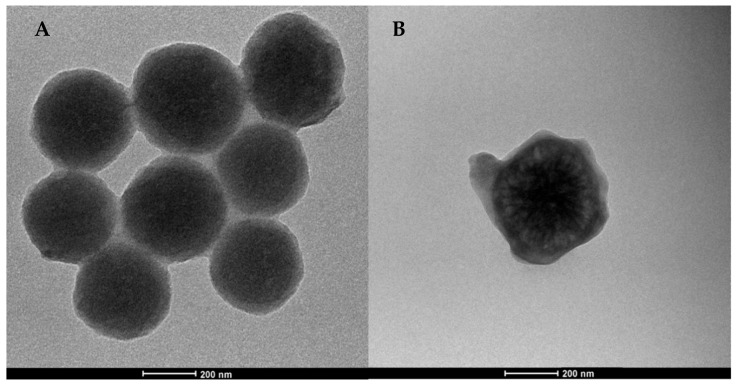
TEM images of PLA/SiO_2__s (**A**) and PLA/SiO_2__w (**B**) nanoparticles.

**Figure 4 nanomaterials-10-00208-f004:**
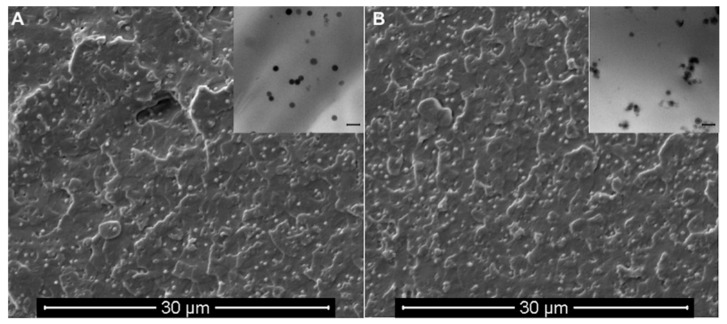
SEM images of PLA/SiO_2__s (**A**) and PLA/SiO_2__w (**B**) nanocomposites; the inserts are the corresponding TEM images of the samples at low magnification (scale bar 400 nm).

**Figure 5 nanomaterials-10-00208-f005:**
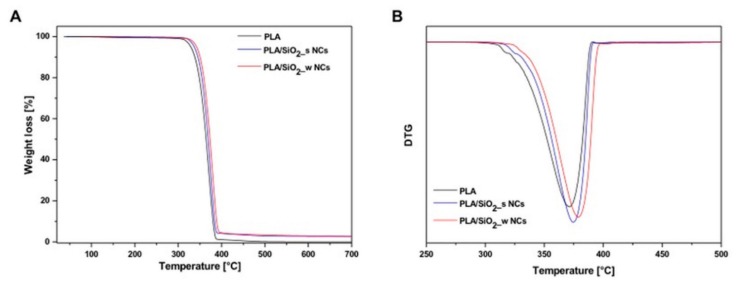
Thermogravimetric (**A**) and Derivative Thermogravimetric (**B**) curves of pure PLA and PLA/SiO_2__s and PLA/SiO_2__w nanocomposites.

**Figure 6 nanomaterials-10-00208-f006:**
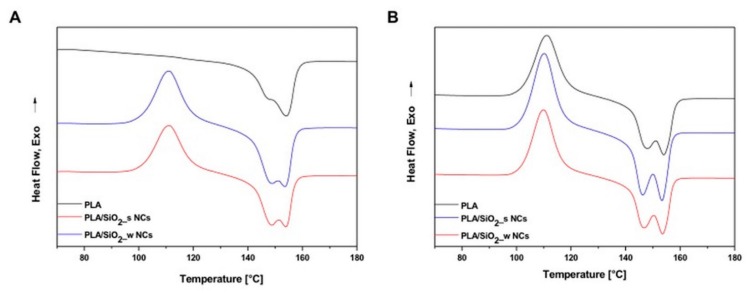
DSC curves of the first (**A**) and second (**B**) heating of pure PLA and PLA/SiO_2__s and PLA/SiO_2__w nanocomposites.

**Figure 7 nanomaterials-10-00208-f007:**
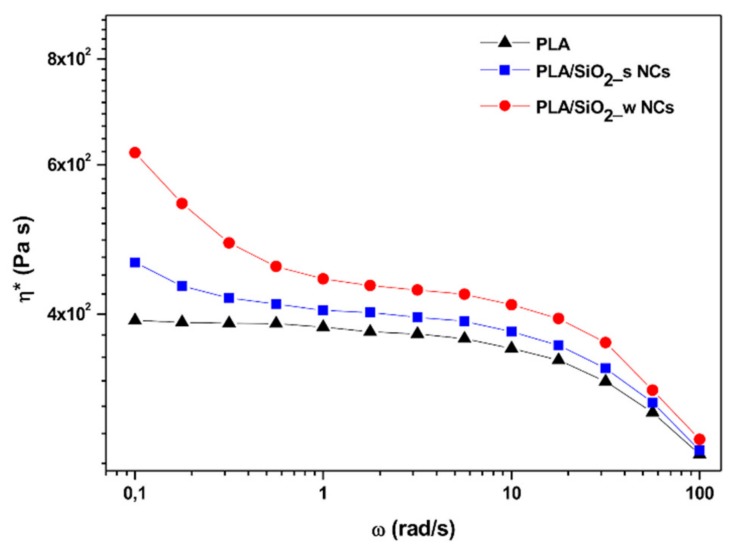
Complex viscosity of pure PLA and PLA/SiO_2__s and PLA/SiO_2__w nanocomposites.

**Figure 8 nanomaterials-10-00208-f008:**
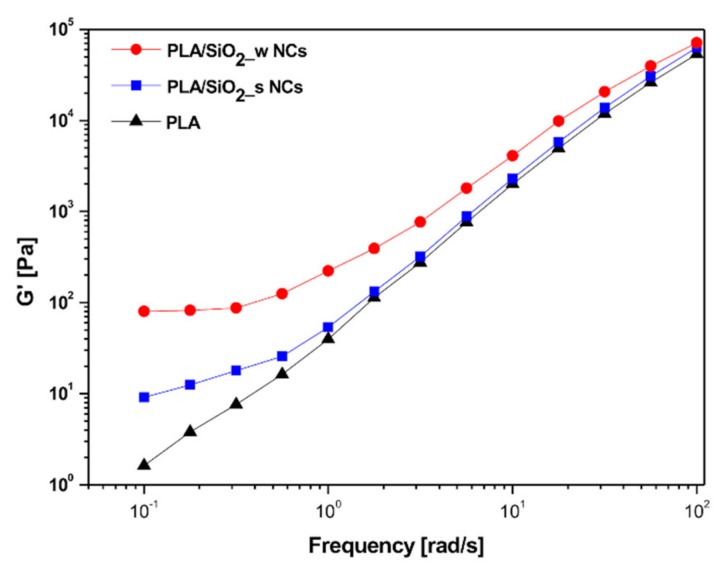
Storage modulus G’ of pure PLA and PLA/SiO_2__s and PLA/SiO_2__w nanocomposites.

**Figure 9 nanomaterials-10-00208-f009:**
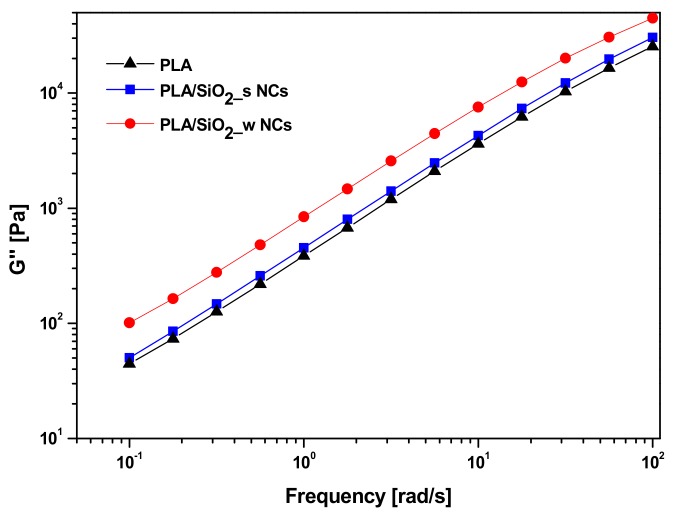
Loss Modulus G’’ of pure PLA and PLA/SiO_2__s and PLA/SiO_2__w nanocomposites.

**Figure 10 nanomaterials-10-00208-f010:**
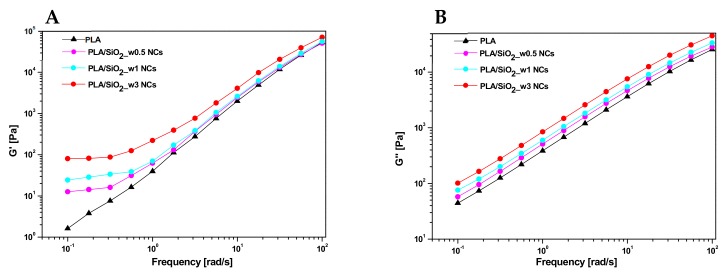
(**A**) Storage modulus and (**B**) loss modulus versus frequency curves at 180 °C for neat PLA and nanocomposites filled with 0.5, 1.0 and 3.0 wt% of wrinkled SiO_2_ nanoparticles.

**Figure 11 nanomaterials-10-00208-f011:**
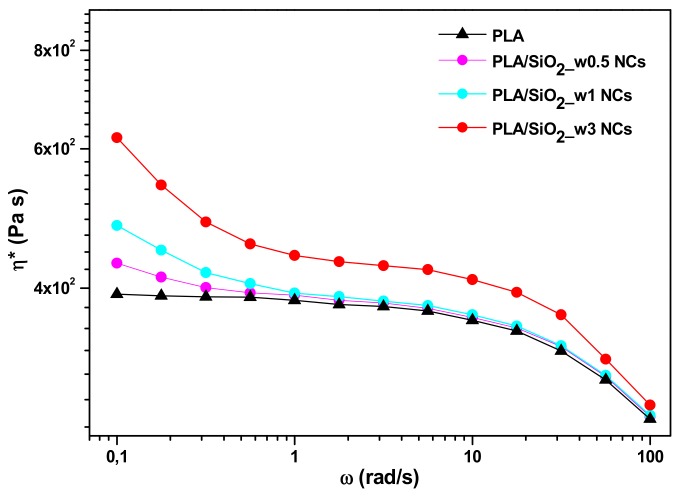
Complex viscosity as a function of frequency at 180 °C for all investigated materials.

**Figure 12 nanomaterials-10-00208-f012:**
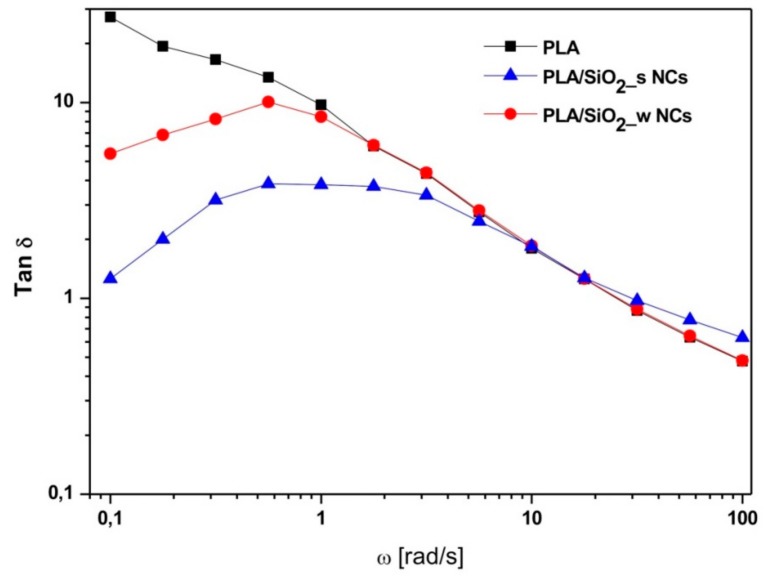
Loss factor tan δ of pure PLA and PLA/SiO_2__s and PLA/SiO_2__w nanocomposites.

**Table 1 nanomaterials-10-00208-t001:** Low-frequency slope of G’ versus ω.

Sample	n
*PLA*	1.320
*PLA/SiO_2__s NCs*	0.602
*PLA/SiO_2__w NCs*	0.074

**Table 2 nanomaterials-10-00208-t002:** Low-frequency (0.01 Hz) rheological parameters.

Sample	G’_0.01Hz_ (Pa)	η*_0.01Hz_ (Pa*s)	λ (s)
*PLA*	1.6	400	40
*PLA/SiO_2__s NCs*	9.1	460	198
*PLA/SiO_2__w NCs*	80	620	1290

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
