# Peer review of "Improving Interaction at Polymer–Filler Interface: The Efficacy of Wrinkle Texture"

_nanomaterials, 2020, doi:10.3390/nano10020208_

Round 1
Reviewer 1 Report
This manuscript presents an effect of wrinkle texture of silica NPs in PLA composites.
The authors reported mechanical properties of a PLA composite using the wrinkle silica NP which was examined in several authors’ papers.
The method of producing nanoparticles has already been published in the authors’ previous papers and was not novel. Although material research on modifying the surface structure of NPs was very interesting, the content of the manuscript was insufficient for an academic paper. For academic papers, it is necessary to present the results of a systematic survey.
Therefore, I believe that this manuscript can be published as a paper in nanomaterials after complete revisions.
I provided major points which should be revised.
(Introduction) The authors should introduce many works to test mechanical properties of polymer nanocomposites filled with NPs with surface modifications using several methods other than the authors. (P4, L148) \lambda was vanished. (P5, L185) I considered a comparison of SAXS profiles between SiO2_s and SiO2_w. Thus, Figure S3 should be moved to main text and added SAXS profile of SiO2_s and discussions about difference of SAXS profiles. I expected a numerical consistency between N2 adsorption/desorption measurements and differences of SAXS profiles can be obtained. (P8, L256) The authors should present the results of G’’, which was addressed in L160 on P4. I recommend to present tan \delta to know the effects of the given wrinkle texture. As a publication of a paper to investigate mechanical properties such as \eta, G’, etc., concentration dependences (especially in the cases of SiO2_w) should be examined because effective surface densities of NP were different between SiO2_s and SiO2_w.
Reviewer 2 Report
The authors studied the effect of surface texture of silica nanoparticles on polymer-filler interactions. The manuscript is well written, but requires major modification before considering for publication.
Author may address following comments:
More results/findings should be added to the Abstract. There are some typos in the text: P.2 L. 75, P.4 L.148, P.5 L.148, … P.2 L. 48-60: Hydrophobic silica particles are now available on the market, which have a good dispersability in hydrophobic media. In addition, due to current regulations, most of dispersing/wetting agents are now non-toxic or in many cases -not classified- at all. Therefore, “avoiding toxic coupling agent” and “hydrophilic nature of silica particles” cannot be considered as convincing motivations for this research work. P.6 L.212: Maximum decomposition temperatures of samples are very close to each other, and do not indicate an improvement in thermal stability. P.7 L. 223-228: DSC results show no clear difference between crystallization behavior of the samples. TGA and DSC curves can be presented as Supplementary Information since they add no important results. Instead of thermal analyses, the authors could have performed simple methods like solvent extraction to determine the polymer bonding layer attached to the surface of the particle. P.9 L.278: High shear forces may destruct particle-particle network, but probably cause no damage to chemical interactions. Do the authors have evidence for damage of chemical interactions at high shear rates? The most important aim of adding nanoparticle, as also pointed out by the authors, is to reinforce the hosting polymer matrix. The results of mechanical analyses such as tensile testing is missing here.Author Response
Please see the attachment.

Round 2
Reviewer 1 Report
This manuscript was a revised manuscript of nanomaterials-679027, which presented an effect of wrinkle texture of silica NPs in PLA composites.
The author has modified the writing of the manuscript, but further revisions are needed for publication.
At least three graphs examining the concentration dependence of SiO2_w presented in the response are essentially necessary to understand what the author claims. These contents should be included in the main text of this manuscript.
Moreover, careful checks for Figures 2 and 10 are required.
Therefore, I believe that this manuscript can be published as a paper in nanomaterials after complete revisions including addition of the results of the concentration dependence of SiO2_w.
I provided major points which should be revised.
(Figure 2) According to Figure 2, the peak position associated with (100) was about 0.5 A^{-1}. It means the corresponding real-space scale was 2 \pi/ 0.5 = 1.25 nm. This result does not support the author's claim quantitatively. There may be a problem with data handling. Careful modification is required.
(Figure 2) It is strongly recommended to check the SAXS profile of SiO2_s. It is important to understand the characteristics of the SAXS profile corresponding to the Wrinkle structure.
(Figure 10) The text font was vanished. Careful checks are required.
Author Response
This manuscript was a revised manuscript of nanomaterials-679027, which presented an effect of wrinkle texture of silica NPs in PLA composites.
The author has modified the writing of the manuscript, but further revisions are needed for publication.
At least three graphs examining the concentration dependence of SiO2_w presented in the response are essentially necessary to understand what the author claims. These contents should be included in the main text of this manuscript. Moreover, careful checks for Figures 2 and 10 are required.
Therefore, I believe that this manuscript can be published as a paper in nanomaterials after complete revisions including addition of the results of the concentration dependence of SiO2_w.
I provided major points which should be revised.
(Figure 2) According to Figure 2, the peak position associated with (100) was about 0.5 A^{-1}. It means the corresponding real-space scale was 2 \pi/ 0.5 = 1.25 nm. This result does not support the author's claim quantitatively. There may be a problem with data handling. Careful modification is required.
-We agree with the reviewer and we are sorry for the mistake, reported x scale of Figure 2 is now corrected.
(Figure 2) It is strongly recommended to check the SAXS profile of SiO2_s. It is important to understand the characteristics of the SAXS profile corresponding to the Wrinkle structure.
-We are sorry, at the moment we are not able to produce this profile. However, we are aware that SAXS spectrum of SiO2_s does not add relevant information. As extensively reported in literature SAXS investigation is usually performed to study very small nanoparticles (2-3 nm) or to investigate the order into porous structure. In the literature a lot of paper reported SAXS investigation on Stöber silica particles. However, these investigations are usually focused on the mechanism of nucleation and growth only in the early stage of the formation of nanoparticles. This type of study is not in the aim of this manuscript.
(Figure 10) The text font was vanished. Careful checks are required.
-We are very sorry. All figures are carefully check to avoid this inconvenient.

Reviewer 2 Report
The authors have addressed my comments.
Round 3
Reviewer 1 Report
I confirmed the revisions of bugs of some texts. This manuscript can be published as a paper in nanomaterials.